# Sequencing and Characterization of *M. morganii* Strain UM869: A Comprehensive Comparative Genomic Analysis of Virulence, Antibiotic Resistance, and Functional Pathways

**DOI:** 10.3390/genes14061279

**Published:** 2023-06-16

**Authors:** Dibyajyoti Uttameswar Behera, Sangita Dixit, Mahendra Gaur, Rukmini Mishra, Rajesh Kumar Sahoo, Maheswata Sahoo, Bijay Kumar Behera, Bharat Bhusan Subudhi, Sutar Suhas Bharat, Enketeswara Subudhi

**Affiliations:** 1Centre for Biotechnology, School of Pharmaceutical Sciences, Siksha ‘O’ Anusandhan (Deemed to be University), Bhubaneswar 751003, Odisha, India; dibya01bioinfo@gmail.com (D.U.B.); sangitadixit2011@gmail.com (S.D.);; 2Drug Development and Analysis Laboratory, School of Pharmaceutical Sciences, Siksha ‘O’ Anusandhan (Deemed to be University), Bhubaneswar 751003, Odisha, India; mahendragaur@soa.ac.in (M.G.);; 3Department of Biotechnology & Food Technology, Punjabi University, Patiala 147002, Punjab, India; 4Department of Botany, School of Applied Sciences, Centurion University of Technology and Management, Bhubaneswar 761211, Odisha, India; 5College of Fisheries, Rani Lakshmi Bai Central Agricultural University, Gwalior Road, Jhansi 284003, Uttar Pradesh, India; beherabk18@yahoo.co.in

**Keywords:** *M. morganii*, pathogens, comparative genomics, AMR, virulence, serotype

## Abstract

*Morganella morganii* is a Gram-negative opportunistic *Enterobacteriaceae* pathogen inherently resistant to colistin. This species causes various clinical and community-acquired infections. This study investigated the virulence factors, resistance mechanisms, functional pathways, and comparative genomic analysis of *M. morganii* strain UM869 with 79 publicly available genomes. The multidrug resistance strain UM869 harbored 65 genes associated with 30 virulence factors, including efflux pump, hemolysin, urease, adherence, toxin, and endotoxin. Additionally, this strain contained 11 genes related to target alteration, antibiotic inactivation, and efflux resistance mechanisms. Further, the comparative genomic study revealed a high genetic relatedness (98.37%) among the genomes, possibly due to the dissemination of genes between adjoining countries. The core proteome of 79 genomes contains the 2692 core, including 2447 single-copy orthologues. Among them, six were associated with resistance to major antibiotic classes manifested through antibiotic target alteration (*PBP3*, *gyrB*) and antibiotic efflux (*kpnH*, *rsmA*, *qacG*; *rsmA*; *CRP*). Similarly, 47 core orthologues were annotated to 27 virulence factors. Moreover, mostly core orthologues were mapped to transporters (*n* = 576), two-component systems (*n* = 148), transcription factors (*n* = 117), ribosomes (*n* = 114), and quorum sensing (*n* = 77). The presence of diversity in serotypes (type 2, 3, 6, 8, and 11) and variation in gene content adds to the pathogenicity, making them more difficult to treat. This study highlights the genetic similarity among the genomes of *M. morganii* and their restricted emergence, mostly in Asian countries, in addition to their growing pathogenicity and resistance. However, steps must be taken to undertake large-scale molecular surveillance and to direct suitable therapeutic interventions.

## 1. Introduction

*Morganella morganii* is a Gram-negative facultative anaerobic, rod-shaped enteric bacterium of the *Enterobacteriaceae* family. It was initially categorized under the *Proteus* genus but subsequently reclassified as a distinct genus through DNA–DNA hybridization analysis [1,2]. This genus is distinguished by its capability to perform trehalose fermentation, generate lysine ornithine decarboxylase and is recognized as the type genus of the newly classified family Morganellaceae [3]. This family comprises eight genera: *Arsenophonus*, *Cosenzaea*, *Moellerella*, *Morganella*, *Photorhabdus*, *Proteus*, *Providencia*, and *Xenorhabdus* [1]. These bacterial species are detected in various ecological niches, including the environment, animals, and human microbiota. This organism is a crucial opportunistic pathogen because it can cause many clinical and community-acquired infections [4]. *M. morganii* has been implicated in various clinical infections, such as urinary tract infections (UTIs) due to long-term urinary catheters, septicemia, and wound infections [5,6], which have more fatal consequences compared to those caused by *Escherichia coli* [7]. Invasive infections caused by *M. morganii* are commonly associated with a considerable mortality rate due to a lack of suitable empirical antibiotic interventions [8]. It has also been associated with various pathological conditions, including brain abscess, liver abscess, chorioamnionitis, peritonitis, pericarditis, septic arthritis, rhabdomyolysis, necrotizing fasciitis following snakebites, bilateral keratitis, neonatal sulfhemoglobinemia, and non-clostridial gas gangrene [8].

*M. morganii* is naturally resistant to ampicillin, amoxicillin, and most of the first- and second-generation cephalosporins due to the presence of the *ampC* resistance gene [4,9,10] as well the last-resort drug, colistin [4]. The use of broad-spectrum antibiotics resulted in the emergence of multidrug-resistant (MDR) or even extensively drug-resistant (XDR) *M. morganii*, leading to the failure of therapy in clinical settings [11]. Various plasmids and transposons or integrons, such as the IncP6 plasmid carrying *bla*_KPC-2_, the IncN plasmid carrying *bla*_OXA-181_, the IncC plasmid carrying *bla*_NDM-1_, the IncX3 plasmid carrying *bla*_NDM-5_, the Tn7 transposon carrying *bla*_IMP-27_, the Tn6741 transposon carrying *bla*_CTX-M-3_, the Tn7 transposon carrying *cfr*, and the In1390 integron carrying *bla*_GES-5_, are also reported to be acquired [12,13,14]. Resistance acquired in *M. morganii* through these integrative and conjugative elements (ICEs) and mobilizable genomic islands (MGIs) poses a clinical treatment challenge [15]. 

Whole genome sequencing of bacteria is the most suitable option for understanding the genetic processes behind this organism’s antibiotic resistance and pathogenicity and monitoring the spread of infections. Oxford nanopore sequencing (ONS) is a nucleic acid sequencing technology that uses protein nanopores to read genome sequences [16,17]. This technology can produce long DNA reads, making it useful for genome assembly, identifying structural variations, and detecting repeats. It has high accuracy in detecting small genetic variations and is essential for identifying antibiotic resistance and virulence in bacteria [17]. ONS also allows real-time sequencing, enabling the rapid identification of pathogens, monitoring bacterial populations, and detecting outbreaks at a reasonable cost within the laboratory setup [18]. 

This study aimed to analyze the genome of *M. morganii* strain UM869, isolated from a urine sample of a patient with a urinary tract infection in Bhubaneswar, India. The research investigated the genome’s virulence factors, drug-resistance mechanisms, prediction of genomic islands, and COGs functional pathways through a comparative genomic analysis using published *M. morganii* genomes.

## 2. Materials and Methods

### 2.1. Identification of Bacteria

The strain UM869 was isolated from a 75-year-old female patient diagnosed with community-acquired urinary tract infections (UTI), who was hospitalized in a super-specialty hospital at Bhubaneswar City, India, in October 2021. The antibiotic susceptibility and species identification of the strain UM869 was performed by an automated VITEK 2 system (bioMerieux, Inc., Hazelwood, Portland, OR, USA). The result was interpreted following CLSI guidelines (CLSI, 2018) [19]. The *E. coli* ATCC 25922 was taken as a reference strain for antimicrobial susceptibility testing analysis. Further, the species of the strain was confirmed by amplification and sequencing of the 16S rRNA.

The strain UM869 was cultured overnight in Luria–Bertani broth medium (LB medium) at 37 °C in a shaking incubator (Remi orbital shaking incubator). The genomic DNA was then extracted from 2 mL of overnight incubated bacterial culture, as described by Sahoo et al. [20]. The quality and quantity of extracted DNA were evaluated using electrophoresis on a 1% agarose gel. The extracted high-quality DNA was subjected to 16S rRNA gene PCR amplification using universal primers [21]. The quality of PCR products was examined using electrophoresis in 1% agarose gel. The PCR-amplified product was outsourced for sequencing at AgriGenome Labs Pvt. Ltd., Cochin, India. The quality-trimmed 16S rRNA sequence was submitted to NCBI’s GeneBank under the accession number ON533444.

### 2.2. Whole Genome Sequencing, De Novo Assembly, and Functional Annotation

The whole genome sequencing of the UM869 strain was accomplished using the Oxford nanopore technology platform at Centurion University of Technology and Management, Bhubaneswar, India. Oxford nanopore technology (ONT) is a sequencing technology that produces long-read sequences (tens of thousands of bases) compared to traditional sequencing methods. These long reads benefit genome assembly, structural variant identification, and repeat detection. ONS is also portable and can be used in remote locations. The flow-cell FLO-MIN106 vR9 containing the prepared genomic DNA library was inserted into the MinION set, followed by sequencing using MinKNOW v1.7.14 [22]. Base-calling was performed using the ONT base-caller Guppy tool [23] and the fastq files were generated from the fast5 file using Poretools [24]. Porechop v0.2.1 [25] was used for adaptor trimming and NanoFilt v2.2.0 [26] was used to remove the reads having quality scores ≤ 20. The QC-passed high-quality long reads were assembled using Flye v2.9 [27] with default parameters, and the assembly files were assessed for quality using QUAST v5.0.2 [28]. The assembled sequences were deposited in the NCBI’s GenBank under the accession CP104700.1. The genome of the UM869 was annotated using the NCBI’s prokaryotic genomes annotation pipeline (PGAP) [29]. The genomic assembly of strain UM869 was explored to identify virulence factors and resistance genes determinants through the Virulence Factors Database (VFDB) and the Comprehensive Antibiotic Resistance Database (CARD), respectively. The genome was further screened for mobile genetic elements (insertion sequence, transposon elements, plasmid signature sequence, and phage elements) using IsFinder [30], TnCentral [31], PlasmidFinder [32], and Phaster [33].

### 2.3. Comparative, Phylogenetic, and Core Orthologues Analysis

The genome assemblies of 81 publicly available *M. morganii* (subspecies *morganii*) were retrieved from the NCBI genome database on November 30, 2022. The completeness, and contamination of all the assemblies were evaluated by CheckM v1.2.2 [34], and BUSCO v5.4.4 [35]. The assembly with completeness ≥90%, and zero contamination (79 genomes) was taken for further downstream analysis.

To evaluate the genetic relatedness among the genomes, average nucleotide identity (ANI) was calculated using the ‘ANI’ module of PGCGAP v1.0.28 [36]. The generated ANI distance matrix was plotted into a heat map using the gplots [37] package in R Studio v4.1.3. The maximum likelihood phylogenetic tree of single-copy core protein was reconstructed using the “CoreTree” module of PGCGAP v1.0.21 [38], and inferences were performed by plotting the tree using iTol [39]. Briefly, the sequence of single-copy core orthologues was extracted using perl scripts [36], and aligned using MAFFT [40], followed by a concatenation of each protein’s alignment. Further, the concatenated alignment of each protein was converted into the corresponding codon alignment using PAL2NAL v14 [41], followed by the calling of core SNPs using SNP sites [42]. Then, a phylogenetic tree was construed based on the best model of evolution using IQ-TREE [43]. OrthoFinder v2.5.4 [44] was used to identify core orthologue proteins among the genomes of *M. morganii* species. The consensus sequence of each core orthologue was generated by multiple sequence alignment using the CIAlign tool [45].

### 2.4. Functional Annotation of Core Orthologues

To annotate the core orthologues, the consensus sequences of all the core orthologues underwent BLASTing against KOfam, which is an HMM database of KEGG orthologues, using kofamKOALA [46] with an e-value threshold of ≥1 × 10^−5^. Subsequently, the eggNOG-mapper tool [47] with the EggNOG database [48] was employed to classify all the core orthologues sequences into clusters of orthologous groups of proteins (COGs). Antimicrobial resistance genes were identified by BLASTing against the Comprehensive Antibiotic Resistance Database (CARD) using RGI v5.1.1 [49,50]. Similarly, the virulence factors were identified by BLASTing core proteins against the Virulence Factors Database (VFDB) [51].

### 2.5. Comparative O-Antigen Gene Cluster (O-AGC) Analysis

The assembled sequences of UM869, and 78 genomes of *M. morganii* were BLAST against serotypes (type 1 to type 11) of *M. morganii* available in the NCBI database [52]. The island map of identified most similar O-AGC was created by gggenes v0.4.1 R package. Additionally, all the serotype genes were clustered at 97% similarity using CD-HIT [53]. Then, the genes were BLAST against the selected serotype sequence. Genes with 90% query coverage and 100% identity were selected to generate the heatmap through OriginPro v2021. A detailed workflow presentation depicting all the steps in the above methodology is shown in Appendix A. 

## 3. Results

### 3.1. Bacterial Identification, and Antibiogram Study

The UM869 strain was isolated from a 75-year-old female patient with urinary tract infections in a super-specialty hospital in Bhubaneswar, Odisha, India. From the VITEK 2 analysis, UM869 was identified as *M. morganii*. The strain showed resistance to major antibiotics such as amoxicillin/clavulanic acid, piperacillin/tazobactam, cefoperazone/sulbactam, cefuroxime, cefepime, imipenem, ertapenem, amikacin, gentamicin, levofloxacin, minocycline, fosfomycin, trimethoprim/sulfamethoxazole,, and colistin. The resistance phenotype was multidrug resistance (MDR), as interpreted using CLSI guidelines [19]. From 16S rRNA gene sequencing, the strain UM869 showed 99.77% identity at 99% query coverage with *M. morganii* NBRC 3848 (accession no. AB680150) through BLASTn analysis [54]. Further, whole genome sequencing using Oxford nanopore technology confirmed the strain as *M. morganii*.

### 3.2. Genome Sequencing of UM869

The de novo assembly of high-quality reads obtained from the nanopore sequencing technology resulted in one contig of 3,761,991 bp size, and GC content of 51%. UM869 had a genome fraction of 49.752%, and a genome completeness, and contamination level of 97.01% and 0.27%, respectively. The genome UM869 (NCBI assembly accession. GCA_025398975) comprised 2870 protein-coding sequences (CDS), 718 pseudogenes, 79 tRNAs, 22 rRNAs, and 1 tmRNA. The *M. morganii* strain UM869 was assembled into a single circular genome.

### 3.3. Resistance Genes, Virulence Factors, and Mobile Genetic Elements of UM869

UM869’s genome comprises 65 genes associated with 30 virulence factors, including efflux pump, hemolysin, urease, serum resistance, iron uptake, adherence factors, toxin, and endotoxin (Appendix A). The genome also contains 11 resistance genes, including *Escherichia coli* EF-Tu mutants, conferring resistance to puromycin; *DHA-17*, *Hemophilus influenzae PBP3,* conferring resistance to β-lactam antibiotics; and *catII* from *Escherichia coli* K-12, *qacG*, *fosA8*, *KpnH*, *rsmA*, *CRP,* and *gyrB*. These genes confer resistance to various classes of antibiotics, including cephalosporins, cephamycins, penams, phenols, macrolides, fluoroquinolones, aminoglycosides, diaminopyrimidines, and phosphonic acid antibiotics (Appendix A). They are also associated with alterations in antibiotic targets (*PBP3*, *gyrB*), the inactivation of antibiotics (*DHA-17*, *fosA8*), and the efflux of antibiotics (*qacG*, *kpnH*, *rsmA*, *CRP*). The genome also contained the insertion sequences IS200G, In36/37, and In6 with sequence identities of 84%, 99%, and 95%, respectively. IS200G is a salmonella-specific insertion sequence and contains the transposon gene (*tnpA*) [55]. This gene was located at position 1,945,287–1,945,977 bp, whereas, In36/37 and In6 were found at positions 580,797–582,017 and 1,049,772–1,050,897 bp, respectively [55]. These two insertion sequences are of *E. coli* plasmid origin (AY259086/5 and L06822) and carry the genes *hypA* (metallo-chaperon), *ampR* (transcriptional activator) and *catA* (chloramphenicol acetyltransferase). However, we obtained no positive results for the plasmid signature sequence and phage elements.

### 3.4. Comparative Phylogenomic Analysis of M. morganii Strains

In this study, we performed a comparative genomics analysis of 82 *M. morganii* genomes retrieved from the NCBI GenBank database “https://www.ncbi.nlm.nih.gov (accessed on 30 November 2022)” from six countries, including the UM869 strain from this study (Appendix A). The completeness and contamination levels of all the genomes ranged from 97.01–100% and 0–8.66%, respectively. UM869 showed 97.01% completeness and 0.27% contamination (Appendix A). From the genome reannotation, the genome size of all strains ranged from 3,618,144 to 4,575,834 bps with varied N50 values (Appendix A). Based on the contamination and completeness of the genomes, we excluded three genomes (GCF_026341575, GCF_018802465, and GCF_003852695) and performed a comparative genomics analysis of the remaining 79 *M. morganii* genomes. 

ANI between strains was calculated and subjected to hierarchical clustering into major groups determined among the 79 genomes to identify the closest strains based on their genome similarities. Genomes with ANI values greater than 95% were considered the same species. Among the strains, the ANI values ranged from 91.79% to 100%, with the highest ANI (100%) observed between GCF_018475065 and GCF_018475585, whereas the lowest (91.79%) was observed between GCF_018456225 and GCF_018475185 (Appendix A). Similarly, the ANI value of the UM869 strain with all other strains ranged between 91.96 and 99.82%. Because the average ANI percentage of all the genomes was 97.92%, which is greater than the ANI cutoff of 95%, all the strains belong to the same species. The ANI tree (Appendix A) is divided into three clusters, namely 1, 2, and 3, containing 64, 2, and 6 strains, respectively. The UM869 strain (GCA_025398975) is clustered (99.82% ANI value) with GCF_018474645.

### 3.5. M. morganii Phylogeny and Genetic Diversity

The core SNP-based phylogenetic analysis exhibited diversity among the genomes of *M. morganii* (Figure 1). Phylogenetic analysis of the 79 *M. morganii* genomes revealed four major clusters and seven singlet nodes, as highlighted in Figure 1. We identified a close phylogenetic relationship between UM869 (GCA_025398975), isolated from urine, and GCF_018474645, isolated from sputum in China. However, it was found that the animal sample, i.e., GCF_018475125 and GCF_018475285, are closely related to the human samples (GCF_018475645 and GCF_018474925), although all four samples are from the same country (China) (Figure 1). The results unequivocally refute the hypothesis that host-specific lineages share a common evolutionary background with the host species under consideration [56]. The close sequence similarity between clinical and zoonotic strains demonstrates that food and the environment significantly transmit the strain from animals to humans and between countries [57].

### 3.6. Identification and Analysis of Orthologues Genes

From the OrthoFinder, we could assign 290,050 genes to 6069 orthogroups, which included 2447 genes belonging to single-copy orthologues, while 1577 genes remained unassigned to any orthogroups. Out of the 6069 orthogroups identified from the 79 genomes, 2692 (44%) were core orthologues (99% ≥ strains ≤ 100%), 306 (5%) were soft-core orthologues (95% ≥ strains < 99%), 1005 (17%) were shell orthologues (15% ≥ strains < 95%), and 2066 (34%) were cloud genes (0% ≥ strains < 15%) as predicted by OrthoFinder. Further, 2692 core orthologous groups were subjected to multiple sequence alignment to extract the consensus sequences for subsequent annotations.

### 3.7. Functional Annotation of Core Genomes

From the functional annotation, 2692 core orthologues were assigned to 7 pathways, 48 super pathways, and 254 sub-pathways (Appendix A). The most commonly identified super pathways in all core orthologues were transporters (468 orthologues), two-component systems (148 orthologues), transcription factors (117 orthologues), ribosomes (114 orthologues), ABC transporters (108 orthologues) with EC numbers (98 orthologues), transfer RNA biogenesis, DNA repair and recombination proteins (88 orthologues), and quorum sensing (77 orthologues) (Appendix A). The top 20 key super pathways (≥15 counts) are shown in Figure 2A. About 95 core orthologues were mapped to the “function unknown” category, suggesting that many aspects of *M. morganii* still require exploration. 

The identified core orthologues of *M. morganii* mapped to 2627 distinct clusters of orthologous groups (COGs) were divided into 21 unique COG categories, as listed in Appendix A. The highest number of COGs (453) belonged to the ‘function unknown’ category [S], followed by [E] amino acid transport and metabolism (269), [K] transcription (225), and [C] energy production and conversion (201). However, the lowest COGs were observed in [A] RNA processing and modification (4), as shown in Figure 2B, while UM869 exhibits 73 cloud orthogroups belonging to 14 COG categories. Out of all COG categories, [S] ‘function unknown’ has the highest number of cloud orthologues (15), followed by [E] amino acid transport and metabolism (9), [P] inorganic ion transport and metabolism (9), and [K] transcription (7). Details of the COG categories with their descriptions are presented in Appendix A. 

### 3.8. Identification of AMR and Virulence Genes

The annotation of core orthologues revealed that multiple antimicrobial resistance genes belong to different resistance mechanisms. Specifically, *KpnH*, *PBP3*, *rsmA*, *CRP*, and *gyrB* genes were identified in all genomes conferring resistance to fluoroquinolone, aminoglycoside, carbapenem, cephalosporin, diaminopyrimidine, phenicol, cephamycin, and macrolide antibiotics, as shown in Table 1. The presence of *qacG* in all the genomes confers resistance to disinfecting agents and antiseptics. Further, four antibiotic efflux resistance mechanisms, including major facilitator superfamily (MFS), small multidrug resistance (SMR), resistance-nodulation-cell division (RND), antibiotic efflux pump, and two antibiotic target alteration resistance mechanisms, were predicted among all the genomes. 

From the VFDB database annotation, only 47 core orthologues annotated to 15 VF classes, 27 virulence factors, and 38 associated genes across all the genomes (Table 2). The most frequently identified virulence factors included the type III secretion system (T3SS), type I fimbriae, endotoxins, and toxins. The secretion system virulence factors class, such as T3SS, T4SS, and TTSS, were found to be particularly prevalent in pathogenic strains of the species, with several genes associated with T3SS, T4SS, and TTSS. Toxins, such as hemolysins, and endotoxins, such as lipooligosaccharide (LOS), were also identified, as were outer-membrane proteins involved in the adhesion and invasion of host cells. Autotransporters and flagella virulence factor classes involved in diverse functions such as adhesion, invasion, toxin secretion, and host colonization were detected less frequently among the strains (Table 2).

### 3.9. Serotype

In this study, we analyzed the serotype content of all 79 strains of *M. morganii,* as reported by Liu et al. [52]. The reported 11 serotypes were BLASTed against all the genomes. In thirty-one genomes, five serotypes were mapped with 100% coverage (Appendix A). The type 8 O-antigen serotype was predicted in the genome UM869 at position 3,495,232 to 3,509,433 bp. The serotype region in UM869 was characterized by ten genes, including *tarF*, *gt1*, *wzy*, *tagD*, *gt2*, *wzx*, *gnu*, *trmL*, *cysE*, and *gpsA,* and five unannotated genes (ORFs) (Figure 3). These reported genes are involved in the biosynthesis and transport of the O-antigen component of the bacterial lipopolysaccharide. The presence of the *wzx* gene, responsible for encoding the O-unit flippase and the *wzy* gene, responsible for encoding the O-antigen polymerase, suggest that *M. morganii* is likely to synthesize its O-antigen via the *wzx*/*wzy*-dependent pathway (Figure 3). In addition, the genes present in all the mapped serotypes were clustered and BLASTed against 31 genomes. The result was visualized by plotting the presence/absence of genes versus the genome, as depicted in Figure 4.

## 4. Discussion

Epidemiological investigations have consistently identified *M. morganii* as a frequent causative agent of nosocomial bacterial infections worldwide [58,59,60,61,62]. Repeated reports on acquired resistance in *M. morganii* reveal more about their life-threatening actions as it further complicates existing treatment options [4]. Despite the serious clinical threat posed by the intrinsic and acquired resistance of *M. morganii*, it has received less attention so far. However, few studies have explored the evolutionary relationships and intricate internal genome structure of multiple genomes of *M. morganii* using genetic information from public databases [3,10,63]. Comparative genomic analysis, combined with a geographical region, isolation source, host, and antibiotic resistance gene content, is valuable for conducting genomic epidemiological analysis. 

In this study, the MDR *M. morganii* UM869 strain genome, obtained from patients with urinary tract infection (UTI), was compared with 78 publicly available *M. morganii* genomes through ANI and core SNP-based phylogenetic analysis. Comparative phylogenomic analysis revealed that the UM869 genome was closely related to the GCF_018474645 (strain FS112720; isolated from sputum) and GCF_018474565 (strain E89; isolated from secretions) genomes from China (Figure 1 and Appendix A). The observed clustering of *M. morganii* strains from different geographic locations and isolation sources in a short timeline over a period from 2015 to 2021 suggests that these strains may be highly clonal [64] and might have spread due to their close geographical location in the map or dissemination due to frequent trade and tourism. 

From the analysis of core orthologues, 290,050 genes were grouped into 6069 orthologues, of which 2692 were core orthologues. These core orthologues proteins usually retain their original function during microorganism evolution and help determine the relationships between genome structure, gene function, and taxonomic classification [65]. However, 2066 cloud genes might reflect the phenotypical traits specific to the group of *M. morganii* [66]. Therefore, it is important to classify these core orthologues into COGs and predict their functions, particularly in emerging pathogens with newly sequenced genomes [65]. This study mapped most core orthologues to transporters and two-component system pathways (Appendix A). These transporters utilize ATP hydrolysis or proton gradient to transport a wide range of substrates across the membrane, including nutrients, toxins, and antibiotics [67,68].

Similarly, 2631 core orthologues were assigned to 2627 distinct COGs belonging to 21 categories. In the UM869 strain, 73 cloud orthologues were predicted and mapped to 14 COG categories. Moreover, most core and cloud orthologues were mapped to the [S] function unknown COG category, which might contain novel functional genes.

Several TCSs have been identified and characterized in *M. morganii*, including the *PhoP*/*PhoQ*, *ArcAB*, *CpxAR*, and *PmrAB* systems responsible for antimicrobial resistance in bacteria. Studies have shown that the *PhoP*/*PhoQ* system regulates phosphate homeostasis, virulence, and antimicrobial resistance [69], suggesting that these strains might be highly resistant to antibiotics. This further complies with the resistance profiling as analyzed through VITEK 2 system. In this study, WGS analysis revealed that UM869 harbored *KpnH*, *qacG*, *rsmA,* and *CRP* efflux pumps, which may confer resistance to most routinely used classes of antibiotics such as macrolide, fluoroquinolone, aminoglycoside, cephalosporin, carbapenem, and colistin antibiotics (Table 1). The *rsmA* gene belongs to the resistance-nodulation-cell division (RND) efflux pump, which regulates quorum sensing, a communication system in bacteria, and the mutated *rsmA* is linked to increased production of biofilm, elastase, and antibiotic resistance [70].

Similarly, *CRP* is a regulatory gene that codes for the cAMP receptor protein, which regulates bacteria’s virulence genes and carbon metabolism [71]. Another multidrug efflux pump *qacG* gene that confers resistance to various antimicrobial agents, identified in Gram-positive and Gram-negative bacteria, is associated with increased resistance to commonly used healthcare disinfectants [72]. Fluoroquinolone resistance could be mediated by a point mutation in *gyrB,* which encodes the β-subunit of DNA gyrase [73]. The point mutation was also observed in *PBP3* (penicillin-binding protein), which results in resistance to β-lactam antibiotics, such as penicillins, cephalosporins, and carbapenems across various bacterial species, including *Escherichia coli*, *Klebsiella pneumoniae*, and *Pseudomonas aeruginosa* [74,75]. However, only the UM869 strain exhibited unique resistant genes such as *fosA8*, *ArnT*, and *qacG*. Previous studies have reported that *fosA8* expression in chromosomal *fosA* genes of *E. coli* significantly confers resistance to fosfomycin [76].

Similarly, *ArnT*, a glycosyltransferase, is essential for bacterial resistance against antimicrobial peptides as it adds 4-amino-4-deoxy-l-arabinose (l-Ara4N) to the lipid A component of lipopolysaccharide, enabling the evasion of antimicrobial effects [77]. The identification of multiple antibiotic resistance mechanisms in *M. morganii* emphasizes the potential threats associated with it. These mechanisms, including efflux pumps and gene mutations, enable bacteria to survive exposure to commonly used antibiotics, complicating treatment and increasing the risk of untreatable infections.

Several virulence factors have been identified in the *M. morganii* genome, including type III secretion system (T3SS), type I fimbriae, endotoxins, and toxins. In the UM869 strain, *fimCDH* genes are predicted as type I fimbriae virulence factors and play an important role in the colonization and pathogenicity of *M. morganii* as these are the most common virulence factors responsible for adherence to surfaces or other cells [4]. The iron acquisition and secretion system (T3SS, T4SS, and TTSS) was the most abundant virulence factor encoded in the UM869 genome that functions in immune evasion (*IgA* protease) and hemolysins [78]. The type-3 secretion system (T3SS) is a highly conserved virulence factor in disease-causing Gram-negative bacteria and is responsible for injecting bacterial effector proteins directly into the host cell cytoplasm [79]. Similarly, iron acquisition genes *sitABCD* mediate manganese–iron transfer and are essential for bacterial survival in iron-deficient environments [80]. The gene *hlyA* was also encoded in the UM869 genome, which is homologous to the α-hemolysin gene of *E. coli*. It binds to the cell surface and matures into a β-barrel transmembrane pore, creating an aqueous channel that permits the transport of small molecules such as K^+^ and Ca^2+^ ions, which causes the necrotic death of the target cell [81]. The UM869 genome also contains the efflux pump (*farB*), which contributes to resistance by pumping out molecules, such as bile salts and antimicrobial peptides, which can help the bacteria, evade the immune system and disease-causing agents [82,83,84]. The mobile genetic elements play a relevant evolutionary role that drives genome plasticity. The insertion sequence and transposon elements in the UM869 genome imply the possibility of disseminating resistant determinants via horizontal gene transfer [85].

The varied structure of O-antigen in bacteria differentiated the *M. morganii* species at the strain level. Liu et al. 2021 [52] developed a molecular serotyping based on diverse O-antigen gene clusters (O-AGC) in *M. morganii*. Based on serotype sequences reported by Liu et al., 2021 [52], 31 genomes were classified into five different serotypes (Figure 4) in our study. The gene structure of serotype cassettes was present in varied combinations in different strains of *M. morganii*, suggesting that the serotyping of *M. morganii* may be complex and require various genotypic and phenotypic methods to be understood [74]. In Gram-negative bacteria, the wzx/wzy-dependent pathway is predominant for producing O-antigen and differentiates the O-antigen clusters [86]. In this study, the serotypes predicted both *wzx* and *wzy* genes in the *M. morganii* genomes.

## 5. Conclusions

This is the first report from India that provides a genomic insight into the diversity and emergence of resistant determinants in *M. morganii* through a comparative genomic study. However, the episodes of the outbreak of *M. morganii* clones are less frequent and restricted to the eastern part of the globe (Asia). Therefore, studying the faster dissemination rate, the acquisition of resistance determinants, and the comprehensive surveillance of the *M. morganii* infection are all highly desirable before this genus potentially causes an uncontrollable epidemic. Further, the enrichment of the *M. morganii* public database will help better understand the bacteria’s origin, evolution, and transmission and also toward designing suitable therapeutics to overcome infections.

## Figures and Tables

**Figure 1 genes-14-01279-f001:**
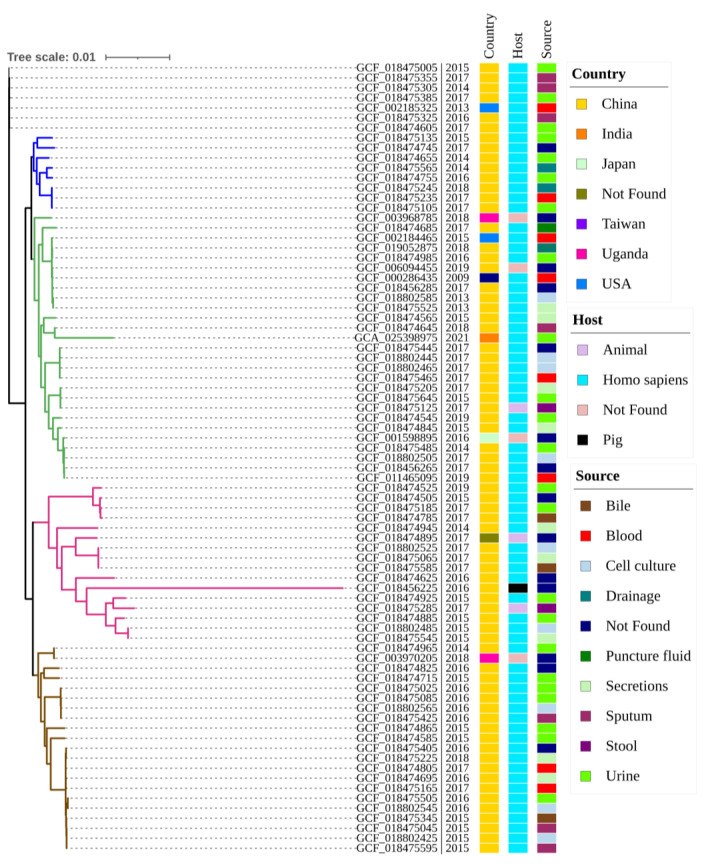
Core SNP-based phylogenetic analysis among 79 strains of *M. morganii*. Metadata such as isolation sources, host, and country are marked with different colors. The year of isolation for each genome is labeled after its accession number. Strains associated with the four clusters are delineated by blue, green, light purple, and light brown branches. The tree scale is presented as the estimated branching length.

**Figure 2 genes-14-01279-f002:**
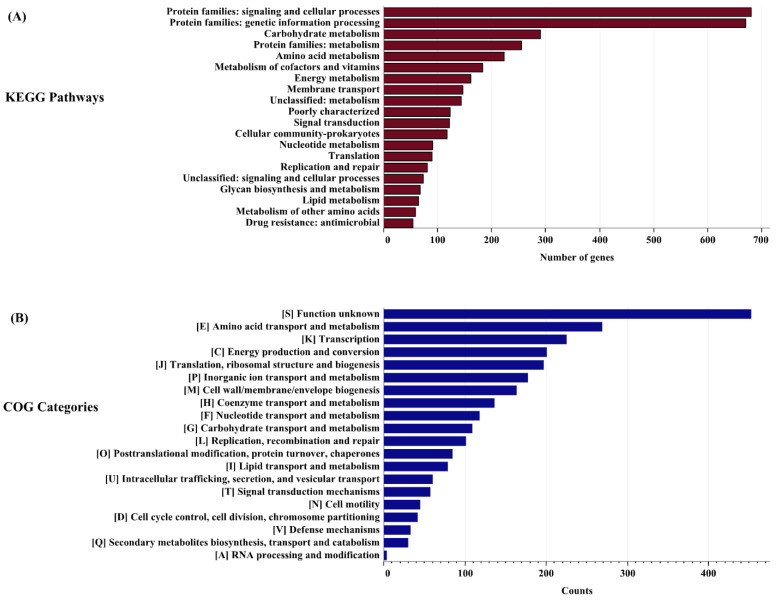
Distribution of core orthologues mapped to KEGG orthologues pathways and clusters of orthologous groups of proteins (COGs). Each bar represents the number of genes in their respective pathways/categories. (**A**) Top 20 KEGG pathways (≥15 counts). (**B**) Distribution of core orthologues in 20 COG categories mapped using eggNOG mapper.

**Figure 3 genes-14-01279-f003:**
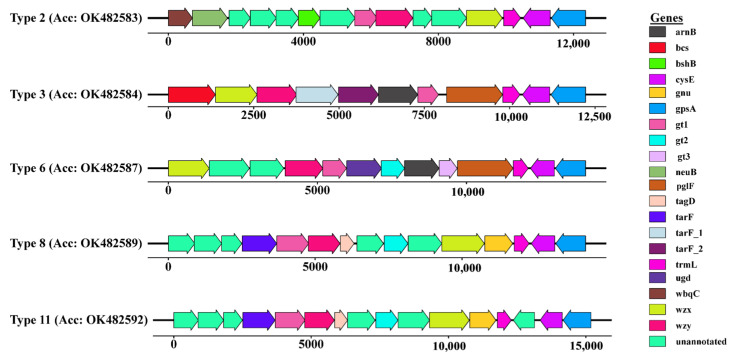
The gene content of serotype cassettes present in 31 out of 79 strains of *M. morganii* was used in this study. The genes and unannotated ORFs are drawn as arrows with orientations (forward and reverse). The image was created with gggenes v0.4.1. The details of mapping are presented in Appendix A.

**Figure 4 genes-14-01279-f004:**
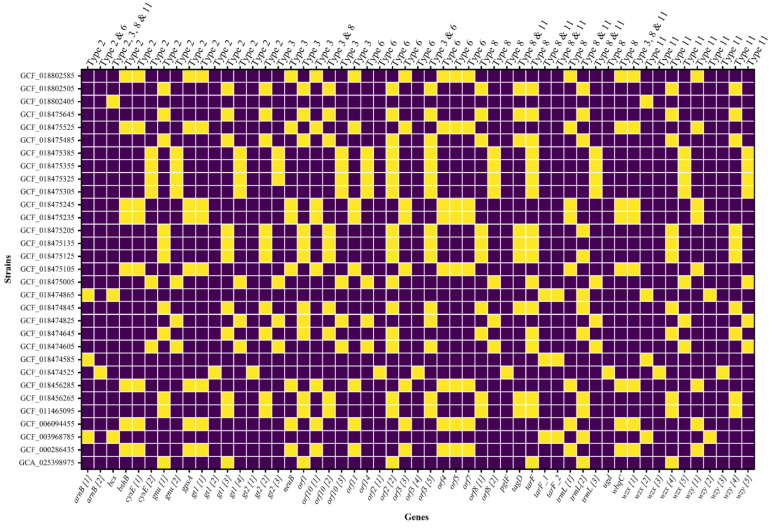
The heatmap illustrates the presence and absence of genes, where yellow indicates gene presence and blue indicates gene absence in their respective strains. The numbers in square brackets represents the cluster variants of the gene at 97% identity. The serotype genes were clustered at 97% similarity using CD-HIT. Genes with query coverage of ≥90% and identity of ≥99% were selected for generating the heatmap.

**Table 1 genes-14-01279-t001:** Antimicrobial resistance genes identified in core orthologues of 79 *M. morganii* strains. The AMR genes were identified using the CARD database in strict mode.

Core Orthologues	Gene	Drug Class	Resistance Mechanism	AMR Gene Family
OG0000319	*KpnH*	Macrolide antibiotic; Fluoroquinolone antibiotic; Aminoglycoside antibiotic; Carbapenem; Cephalosporin; Penam; Peptide antibiotic; Penem	Antibiotic efflux	Major facilitator superfamily (MFS) antibiotic efflux pump
OG0000873	*PBP3*	Cephalosporin; Cephamycin; Penam	Antibiotic target alteration	Penicillin-binding protein mutations conferring resistance to β-lactam antibiotics
OG0001323	*qacG*	Disinfecting agents and antiseptics	Antibiotic efflux	Small multidrug resistance (SMR) antibiotic efflux pump
OG0002043	*rsmA*	Fluoroquinolone antibiotic; Diaminopyrimidine antibiotic; Phenicol antibiotic	Antibiotic efflux	Resistance-nodulation-cell division (RND) antibiotic efflux pump
OG0002548	*CRP*	Macrolide antibiotic; Fluoroquinolone antibiotic; Penam	Antibiotic efflux	Resistance-nodulation-cell division (RND) antibiotic efflux pump
OG0002685	*gyrB*	Fluoroquinolone antibiotic	Antibiotic target alteration	Fluoroquinolone-resistant *gyrB*

**Table 2 genes-14-01279-t002:** Details of identified putative virulence factors in the core orthologues of 79 *M. morganii* strains.

Core Orthologues	Virulence Gene	Virulence Factors	VF Class
OG0000167	*fimD*	Type I fimbriae	Adherence
OG0000540			
OG0001077			
OG0002806			
OG0001650	*cheB*	Flagella (Burkholderia)	Autotransporter
OG0001651	*cheR*		
OG0001256	*chuS*	Heme uptake	Iron uptake
OG0001254	*chuU*		
OG0000956	*ireA*	Iron-regulated element	
OG0000280	*sitA*	Iron/manganese transport	
OG0001230	*sitB*		
OG0001229	*sitC*		
OG0000279	*sitD*		
OG0000533	*basG*	Acinetobactin (Acinetobacter)	
OG0002794	*feoA*	Ferrous iron transport (Legionella)	
OG0002449	*hemG*	Heme biosynthesis (Hemophilus)	
OG0001080	*phoQ*	PhoPQ (Salmonella)	Regulation
OG0000220	*spaP*	Bsa T3SS (Burkholderia)	Secretion system
OG0000295	*flhB*	Flagella (cluster I)	
OG0000933	*exsA*	T3SS (Aeromonas)	
OG0000857	*-*	T4SS effectors (Coxiella)	
OG0000142	*invC*	TTSS (SPI-1 encode)	
OG0000221	*ysaS*	Ysa TTSS (Yersinia)	
OG0000109	*ysaV*	Ysa TTSS (Yersinia)	
OG0000422	*hlyA*	HemolysinHlyA (Aeromonas)	Toxin
OG0000833	*farB*	FarAB (Neisseria)	Efflux pump
OG0001671	*htrB*	LOS (Hemophilus)	Endotoxin
OG0002647	*lgtF*		
OG0002313	*lpxA*		
OG0000783	*lpxH*		
OG0001828	*lpxK*		
OG0002643	*opsX/rfaC*		
OG0000370	*wecA*		
OG0001781	*fimC*	Fim (Salmonella)	Fimbrial adherence determinants
OG0001782	*fimD*		
OG0001783	*fimH*		
OG0001347	*-*	Capsule (Acinetobacter)	Immune evasion
OG0001468	*mgtB*	Mg^2+^ transport (Salmonella)	Magnesium uptake
OG0001659	*mgtC*		
OG0001656	*motA*	Flagella (Bordetella)	Motility
OG0001655	*motB*		
OG0002110	*-*	Cysteine acquisition	
OG0001526	*msbB2*	MsbB2 (Shigella)	Others
OG0000371	*-*	O-antigen (Yersinia)	
OG0000253	*galE*		
OG0000285	*-*	LPS rfb locus	Serum resistance
OG0001161	*katA*	Catalase	Stress adaptation

## Data Availability

WGS sequence reads were submitted to the NCBI’s Bioproject database with the accession ID: PRJNA598939.

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
