# Peer review of "Sequencing and Characterization of *M. morganii* Strain UM869: A Comprehensive Comparative Genomic Analysis of Virulence, Antibiotic Resistance, and Functional Pathways"

_genes, 2023, doi:10.3390/genes14061279_

Round 1

Reviewer 1 Report

Title: The title could be improved by modifying it to "Sequencing and Characterization of M. morganii Strain UM869: A Comprehensive Comparative Genomic Analysis of Virulence, Antibiotic Resistance, and Functional Pathways." This change emphasizes both the characterization of the novel strain and the comparison with all available strains, making it clear that the paper contains information about the new strain as well as the results from the comparative genomics analysis.

Abstract: To improve clarity and structure, please consider rewriting the abstract to align with the structure presented in the results section. This will help readers better understand the study's purpose, methods, and findings.

Line 59: The syntax of this sentence is unclear. Please consider rewriting it to ensure proper grammar and readability.

Line 73: Instead of "genetic codes," the correct term should be "genome sequences." Please revise accordingly.

Line 166: In this paragraph, include the number of assembled contigs. If a single circular contig was assembled using Nanopore reads, please mention it.

Line 173: Consider adding a table with three columns: antibiotic, phenotypic resistance (I/S/R), and the presence of the corresponding resistance gene (Present/Absent). This will help readers understand the relationship between antibiotic resistance genes and phenotypic resistance. Although a GWAS approach is outside the scope of this study, it is worth mentioning.

Methods: The bioinformatic pipeline is excellent.

Figure 1: The color legend for Source metadata is difficult to distinguish due to similar colors. Please improve the color scheme. If the goal is to discuss relationships with metadata, use a cladogram. For presenting the genetic diversity of the new strain compared to others, use a non-circular phylogram.

Line 220: Since this information has been demonstrated in previous studies, please add a citation to the discussion: https://bmcgenomics.biomedcentral.com/articles/10.1186/s12864-020-07001-2

Line 221: This hypothesis is unclear. Please rewrite it and provide supporting evidence from the literature.

Figure 2: This figure may not be necessary, as the four numbers can be included in the text or a small table instead.

Figure 3: The current presentation of the top KEGG pathways is not optimal for several reasons. First, the pie chart is confusing due to the high number of classes. Second, it is known that some genes belong to multiple pathways, some of which may be irrelevant to the species of interest (e.g., Cancer: overview, Aging). Consider revising the visualization for better clarity and relevance. Instead of a pie chart, consider using a horizontal bar chart to illustrate the distribution of the top 31 KEGG pathways (≥ 15 counts) linked to the consensus sequence of 80 M. morganii genomes. Each bar will represent the number of genes detected in the respective pathway, with the length of the bar corresponding to the gene count. The pathways can be ordered from highest to lowest gene count for better readability and comprehension.

Line 286: Consider representing these data as a heatmap, with isolates on the Y-axis and serotype genes on the X-axis. This visualization could help reveal new patterns that might not be visible by simply listing the genes.

Figure 4: It might be useful to merge Figure 4 with Figure 3 into a two-panel figure (A, B).

Line 311: Please add references for all comparative genomics studies of M. morganii.

Line 319: Clarify the term "clustering" and improve the visibility of the tree. This hypothesis is extremely interesting, but it would be more impactful if the tree were more clearly presented.

Line 328: Rewrite this sentence to emphasize that cloud genes often encode strain-specific functions or represent recent horizontal gene transfer events, making them particularly interesting to study. If cloud genes specific to the newly sequenced strain were found, include these results in the COGs section and discuss their implications.

Line 341: Consider removing the discussion about amino acid transport and metabolism, as it may simply indicate that this is a conserved pathway.

Line 360: Please provide a common conclusion regarding the results about antibiotic resistance, as the current text seems to be a mere reiteration of the results with some literature references. Also, discuss the resistance results specifically for the newly sequenced strain.

Line 379: Rewrite gene names in italics for proper formatting.

Author Response

Paper Title: Sequencing and Characterization of M. morganii Strain UM869: A Comprehensive Comparative Genomic Analysis of Virulence, Antibiotic Resistance and Functional Pathways (Manuscript Number: GENES-2331565).

Dear Editor,

We are grateful to you for considering our paper titled “Sequencing and Characterization of M. morganii Strain UM869: A Comprehensive Comparative Genomic Analysis of Virulence, Antibiotic Resistance and Functional Pathways” for revision for your prestigious journal. The reviewer has raised some very interesting and critical points which has been addressed in the manuscript. We have now made all the necessary changes as requested by the reviewer.

Responces to the Reviewer

  1. Title: The title could be improved by modifying it to "Sequencing and Characterization of M. morganii Strain UM869: A Comprehensive Comparative Genomic Analysis of Virulence, Antibiotic Resistance, and Functional Pathways." This change emphasizes both the characterization of the novel strain and the comparison with all available strains, making it clear that the paper contains information about the new strain as well as the results from the comparative genomics analysis.

Response: Thank you for changing the title. We agree with this comment and modified the title accordingly.

  1. Abstract: To improve clarity and structure, please consider rewriting the abstract to align with the structure presented in the results section. This will help readers better understand the study's purpose, methods, and findings.

Response: Thank you for this suggestion. Now we have modified the abstract at line number 21-39.

  1. Line 59: The syntax of this sentence is unclear. Please consider rewriting it to ensure proper grammar and readability.

Response: As per the suggestion we have modify the sentence. “M. morganii is naturally resistant to ampicillin, amoxicillin, and most of the first- and second-generation cephalosporins due to the presence of the ampC resistance gene” at the line number 63-64.

  1. Line 73: Instead of "genetic codes," the correct term should be "genome sequences." Please revise accordingly.

Response: We have added the suggested content to the manuscript on line number 78.

  1. Line 166: In this paragraph, include the number of assembled contigs. If a single circular contig was assembled using Nanopore reads, please mention it.

Response: The sentence has been updated UM869 was assembled into a single circular genome” at line number 188.

  1. Line 173: Consider adding a table with three columns: antibiotic, phenotypic resistance (I/S/R), and the presence of the corresponding resistance gene (Present/Absent). This will help readers understand the relationship between antibiotic resistance genes and phenotypic resistance. Although a GWAS approach is outside the scope of this study, it is worth mentioning.

Response: We think this is an excellent suggestion. Now we have modified the table S1B.

  1. Methods: The bioinformatic pipeline is excellent.

Response: Thank you for the appreciation.

  1. Figure 1: The color legend for Source metadata is difficult to distinguish due to similar colors. Please improve the color scheme. If the goal is to discuss relationships with metadata, use a cladogram. For presenting the genetic diversity of the new strain compared to others, use a non-circular phylogram.

Response: As per the reviewer's suggestions, we have changed the color combination in the Figure 1.

  1. Line 220: Since this information has been demonstrated in previous studies, please add a citation to the discussion:https://bmcgenomics.biomedcentral.com/articles/10.1186/s12864-020-07001-2

Response:  Agree. We have added the suggested citation accordingly at the line number 246.

  1. Line 221: This hypothesis is unclear. Please rewrite it and provide supporting evidence from the literature.

Response: The Line no. 221 has been modified as per reviewer’s suggestion “The close sequence similarity between clinical and zoonotic isolates demonstrates that food and the environment play a significant role in the transmission of the isolate from animals to humans and between the countries (Sahoo et al., 2023)” now at the line number 245-249.

  1. Figure 2: This figure may not be necessary, as the four numbers can be included in the text or a small table instead.

Response: As per the reviewer's suggestions, we remove the figure 2 from the manuscript.

  1. Figure 3: The current presentation of the top KEGG pathways is not optimal for several reasons. First, the pie chart is confusing due to the high number of classes. Second, it is known that some genes belong to multiple pathways, some of which may be irrelevant to the species of interest (e.g., Cancer: overview, Aging). Consider revising the visualization for better clarity and relevance. Instead of a pie chart, consider using a horizontal bar chart to illustrate the distribution of the top 31 KEGG pathways (≥ 15 counts) linked to the consensus sequence of 80 M. morganii genomes. Each bar will represent the number of genes detected in the respective pathway, with the length of the bar corresponding to the gene count. The pathways can be ordered from highest to lowest gene count for better readability and comprehension.

Response: Thank you for the suggestion. Now figure 3 has been updated and merged it into figure 4. The new figure number is Figure 2A and B.

  1. Line 286: Consider representing these data as a heatmap, with isolates on the Y-axis and serotype genes on the X-axis. This visualization could help reveal new patterns that might not be visible by simply listing the genes.

Response: As per reviewers’ suggestion we have added another heatmap figure for serotype gene data as Figure 4.  We kept the original serotype figure as it shows the genes content of the mapped serotypes as well as the length and the direction (forward/reverse). We also cited the figure in the result at line number 327.

  1. Figure 4: It might be useful to merge Figure 4 with Figure 3 into a two-panel figure (A, B).

Response: As per reviewers’ suggestion we merged Figure 3 and figure 4 into one figure as figure 2A and 2B.

  1. Line 311: Please add references for all comparative genomics studies of M. morganii.

Response: Thank you for the suggestion. Now we have added the references at line number 348.

  1. Line 319: Clarify the term "clustering" and improve the visibility of the tree. This hypothesis is extremely interesting, but it would be more impactful if the tree were more clearly presented.

Response: Here “clustering” means in same clad of the tree, which represent the similarity of the isolates. Tree clustering was shown in different color in the phylogenetic tree.

  1. Line 328: Rewrite this sentence to emphasize that cloud genes often encode strain-specific functions or represent recent horizontal gene transfer events, making them particularly interesting to study. If cloud genes specific to the newly sequenced strain were found, include these results in the COGs section and discuss their implications.

Response: Revised the COG results according to the reviewer suggestion and is present at the line number 274 to 279 & 373 to 376.

  1. Line 341: Consider removing the discussion about amino acid transport and metabolism, as it may simply indicate that this is a conserved pathway.

Response: As per the suggestion we have removed the amino acid transport and metabolism in the discussion part.

  1. Line 360: Please provide a common conclusion regarding the results about antibiotic resistance, as the current text seems to be a mere reiteration of the results with some literature references. Also, discuss the resistance results specifically for the newly sequenced strain.

Response: As per the reviewer's suggestions, we have added the common conclusion  and discussion regarding antibiotic resistance at line no 403-407.

  1. Line 379: Rewrite gene names in italics for proper formatting.

Response: Revised accordingly across the manuscript.

Reviewer 2 Report

In this study a clinical isolate of Morganella morganii is isolated and characterized. The genome of the strain is compared to 79 other Morganella morganii genomes that have been deposited to sequence databases. The subject matter is topical and interesting owing to the increased frequency of M. morganii infection. However, there are some points that the authors need to address. Furthermore, this manuscript would benefit from a thorough proofread by a native English speaker to improve grammar.

 I have the following comments for the authors

In the introduction section the authors, discussed mobile genomic islands carrying antimicrobial resistance and stated that these islands pose a challenge to clinical treatment and, according to the authors, one of the research objectives was to investigate genomic islands. In addition, in the abstract the authors stated that the high similarity of the genomes might be due to dissemination of genes via horizontal gene transfer. Furthermore, in the conclusion section, the authors also mentioned that M. morganni has the potential to become epidemic due to its high rate of dissemination and the acquisition of resistance and virulence factors through genetic transfer. However, the authors did not characterize horizontal gene transfer determinants such as mobile genetic elements and their association with amr and virulence. Why were mobile genetic elements not characterized?

Materials and methods

How was drug susceptibility interpreted? Which reference strains were used?

Figure 1. Looking at the Figure it is difficult to identify the four clusters that the authors are referring to. I would recommend that the strains belonging to the four clusters be colour coded. This will make the Figure easy to follow. Could the authors also indicate the 17 branches that they are referring to in the Figure?

Figure 2. Colors are not very contrasting enough which makes it difficult to follow the proportion of each gene category

The proper format for citations of β-lactamase genes should be followed throughout the manuscript i.e., italicized bla, followed by subscript allele designation.

Bacteria names should be italicized throughout the manuscript e.g., Line 40, 45, 47-48 etc.

Table S2: %GC values not included in Table

Line 22-25 “The resistance to major important classes of antibiotics manifested in these genomes is deciphered to be through antibiotic target alteration (PBP3, gyrB), antibiotic inactivation (DHA-17, FosA8) and antibiotic efflux (qacG, kpnH, rsmA, CRP) which directs complicated treatment procedures.” I would recommend that the authors should revise this sentence to “The resistance to major important classes of antibiotics manifested in these genomes is deciphered to be through antibiotic target alteration (PBP3, gyrB), antibiotic inactivation (DHA-17, FosA8) and antibiotic efflux (qacG, kpnH, rsmA, CRP) which complicates antibiotic therapy.”

Line 27-28 “Twenty-eight virulence factors and 36 associated genes belonging to type III secretion system (T3SS), type I fimbriae, endotoxins.” This sentence is incomplete

Line 59-60 “The presence of intrinsic AmpC resistance gene, M. morganii is naturally resistant to

ampicillin, amoxicillin, and most of the first- and second-generation cephalosporins” revise to “M. morganii is naturally resistant to ampicillin, amoxicillin, and most of the first- and second-generation cephalosporins due to the presence of the ampC resistance gene”

Line 39 “Morganella morganii, a Gram-negative facultative anaerobic rod-shaped enteric bacterium of Enterobacteriaceae family” revise to “Morganella morganii, is a Gram-negative facultative anaerobic rod-shaped enteric bacterium of the Enterobacteriaceae family”

Line 78 “with in” revise to “within”

Line 121 “resistance genes determinates” revise to “resistance genes determinants”

Line 179 “These genes are resistance to cephalosporin ……”  revise to “These genes confer resistance to cephalosporins….”

Line 179-183 “These genes are resistance to cephalosporin, cephamycin, penam, phenicol, macrolide, fluoroquinolone, aminoglycoside, diaminopyrimidine, phosphonic acid antibiotic and disinfecting agents/ antiseptics (Table S1B) having antibiotic target alteration (PBP3, gyrB), antibiotic inactivation (DHA-17, FosA8) and antibiotic efflux (qacG, kpnH, rsmA, CRP)” This sentence is not clear need to be rephrased

Line 191 “36,18, 144 to 45,75, 834” revise to 3,618,144 to 4,575,834

Line 213 “Four clear clustering and 17 phylogenetic branches were shown in the Figure 1 comprising 79 M. morganii genomes” revise to “Phylogenetic analysis of the 79 M. morganii genomes revealed four clusters and 17 phylogenetic branches”

Line 267 “Also, we found penem drugs, which belong to the unsaturated” sentence not clear; do the authors mean penem encoding genes?

Line 269 “Further, four antibiotic efflux resistance mechanisms having major facilitator superfamily (MFS)…” revise to “Further, four antibiotic efflux resistance mechanisms including major facilitator superfamily (MFS)…”

Line 289 “34,95,232 to 35,09,433 bp” revise to “3,495,232 to 3,509,433 bp”

Line 322 “spread due their…” revise to “spread due to their…”

Line 361 “Several virulence factors have been identified M. morganii genome” revise to “361 “Several virulence factors have been identified in M. morganii genome”

Author Response

Paper Title: Sequencing and Characterization of M. morganii Strain UM869: A Comprehensive Comparative Genomic Analysis of Virulence, Antibiotic Resistance and Functional Pathways (Manuscript Number: GENES-2331565).

Dear Editor

We are grateful to you for considering our paper titled “Sequencing and Characterization of M. morganii Strain UM869: A Comprehensive Comparative Genomic Analysis of Virulence, Antibiotic Resistance and Functional Pathways” for revision for your prestigious journal. The reviewer has raised some very interesting and critical points which has been addressed in the manuscript. We have now made all the necessary changes as requested by the reviewer.

Responses to the Reviewer

In this study a clinical isolate of Morganella morganii is isolated and characterized. The genome of the strain is compared to 79 other Morganella morganii genomes that have been deposited to sequence databases. The subject matter is topical and interesting owing to the increased frequency of M. morganii infection. However, there are some points that the authors need to address. Furthermore, this manuscript would benefit from a thorough proofread by a native English speaker to improve grammar.

 I have the following comments for the authors

  1. In the introduction section the authors, discussed mobile genomic islands carrying antimicrobial resistance and stated that these islands pose a challenge to clinical treatment and, according to the authors, one of the research objectives was to investigate genomic islands. In addition, in the abstract the authors stated that the high similarity of the genomes might be due to dissemination of genes via horizontal gene transfer. Furthermore, in the conclusion section, the authors also mentioned that M. morganni has the potential to become epidemic due to its high rate of dissemination and the acquisition of resistance and virulence factors through genetic transfer. However, the authors did not characterize horizontal gene transfer determinants such as mobile genetic elements and their association with amr and virulence. Why were mobile genetic elements not characterized?

Response: We agree to this point that mobile genetic elements need to be characterized. In response to this, we have used ISfinder and TnCentral and could detect Salmonaella specific insertion sequence (IS200G) and In36/37 and In6 of E. coli plasmid origin in M. morganni UM869 genome. We also tried to find plasmid and phage in the sequence of our isolate M. morganni UM869 using PlasmidFinder and Phaster but failed to get any positive result. The short methodology of detection and respective results in details is presented in place at line number 130-132, 200-207, 427-429.

Materials and methods

  1. How was drug susceptibility interpreted? Which reference strains were used?

Response: The result was interpreted as per CLSI guideline (CLSI, 2020). The E.coli ATCC 25922 was taken as a reference strain for antimicrobial susceptibility testing analysis. This sentence has been incorporated into the materials and methods section of the manuscript at the line number 98 to 100.

  1. Figure 1. Looking at the Figure it is difficult to identify the four clusters that the authors are referring to. I would recommend that the strains belonging to the four clusters be color coded. This will make the Figure easy to follow. Could the authors also indicate the 17 branches that they are referring to in the Figure?

Response: Now we modified the figure as per reviewer’s suggestion.

  1. Figure 2. Colors are not very contrasting enough which makes it difficult to follow the proportion of each gene category

Response: As per the 1st reviewer's suggestion, we remove the figure 2 from the manuscript and the predicted gene counts were mentioned in the manuscript at the line number 251-258.

  1. The proper format for citations of β-lactamase genes should be followed throughout the manuscript i.e., italicized bla, followed by subscript allele designation.

Response: Thank you. We have modified the citations of β-lactamase genes throughout the manuscript.

  1. Bacteria names should be italicized throughout the manuscript e.g., Line 40, 45, 47-48 etc.

Response: Revised accordingly.

  1. Table S2: %GC values not included in Table

Response: Thank you! Now the %GC was added in the Table S2.

  1. Line 22-25 “The resistance to major important classes of antibiotics manifested in these genomes is deciphered to be through antibiotic target alteration (PBP3, gyrB), antibiotic inactivation (DHA-17, FosA8) and antibiotic efflux (qacG, kpnH, rsmA, CRP) which directs complicated treatment procedures.” I would recommend that the authors should revise this sentence to “The resistance to major important classes of antibiotics manifested in these genomes is deciphered to be through antibiotic target alteration (PBP3, gyrB), antibiotic inactivation (DHA-17, FosA8) and antibiotic efflux (qacG, kpnH, rsmA, CRP) which complicates antibiotic therapy.”

Response: Thank you. Now, we have restructured the abstract as per the 1st reviewer suggestion.

  1. Line 27-28 “Twenty-eight virulence factors and 36 associated genes belonging to type III secretion system (T3SS), type I fimbriae, endotoxins.” This sentence is incomplete

Response: Now we have restructured the abstract as per the 1st reviewer suggestion.

  1. Line 59-60 “The presence of intrinsic AmpC resistance gene, M. morganii is naturally resistant to ampicillin, amoxicillin, and most of the first- and second-generation cephalosporins” revise to “M. morganii is naturally resistant to ampicillin, amoxicillin, and most of the first- and second-generation cephalosporins due to the presence of the ampC resistance gene”

Response: As per the suggestion the sentence was modified at line number 64-66.

  1. Line 39Morganella morganii, a Gram-negative facultative anaerobic rod-shaped enteric bacterium of Enterobacteriaceae family” revise to “Morganella morganii, is a Gram-negative facultative anaerobic rod-shaped enteric bacterium of the Enterobacteriaceae family”

Response: We have added the suggested content to the manuscript on line number 45-46.

  1. Line 78 “with in” revise to “within”

Response: We change the word “with in” to “within” at line number 85.

  1. Line 121 “resistance genes determinates” revise to “resistance genes determinants”

Response: Revised accordingly at the line number 129.

  1. Line 179 “These genes are resistance to cephalosporin ……” revise to “These genes confer resistance to cephalosporins….”

Response: We have added the suggested content to the manuscript.

  1. Line 179-183 “These genes are resistance to cephalosporin, cephamycin, penam, phenicol, macrolide, fluoroquinolone, aminoglycoside, diaminopyrimidine, phosphonic acid antibiotic and disinfecting agents/ antiseptics (Table S1B) having antibiotic target alteration (PBP3, gyrB), antibiotic inactivation (DHA-17, FosA8) and antibiotic efflux (qacG, kpnH, rsmA, CRP)” This sentence is not clear need to be rephrased

Response: As per the suggestion the sentence is now rephrased asThese genes confer resistance to various classes of antibiotics including cephalosporins, cephamycins, penams, phenicols, macrolides, fluoroquinolones, aminoglycosides, diaminopyrimidines, and phosphonic acid antibiotics. They are also associated with alterations in antibiotic targets (PBP3, gyrB), inactivation of antibiotics (DHA-17, FosA8), and efflux of antibiotics (qacG, kpnH, rsmA, CRP)” at line number 198-200.

  1. Line 191 “36,18, 144 to 45,75, 834” revise to 3,618,144 to 4,575,834

Response: We have revised the number format.

  1. Line 213 “Four clear clustering and 17 phylogenetic branches were shown in the Figure 1 comprising 79 M. morganii genomes” revise to “Phylogenetic analysis of the 79 M. morganii genomes revealed four clusters and 17 phylogenetic branches”

Response: We have revised the line “Phylogenetic analysis of the 79 M. morganii genomes revealed 4 major clusters and 7 singlet node which comprises 11 phylogenetic branches as highlighted in Figure 1” at line number 238 to 240.

  1. Line 267 “Also, we found penem drugs, which belong to the unsaturated” sentence not clear; do the authors mean penem encoding genes?

Response: In this sentence, we are attempting to convey that the resistance determinants discovered in the M. morganni UM869 strain confirm resistance to antibiotics belonging to the penem group (such as meropenem, imipenem, etc.). This is also mentioned between lines 276 and 280. As a result, now the sentence is “Specifically, KpnH, PBP3, rsmA, CRP, and gyrB genes were identified in all genome conferring resistance to fluoroquinolone, aminoglycoside, carbapenem, cephalosporin, diaminopyrimidine, phenicol, cephamycin and macrolide antibiotics shown in Table 1.” And the sentence about penem drug is omitted from the manuscript.

  1. Line 269 “Further, four antibiotic efflux resistance mechanisms having major facilitator superfamily (MFS)…” revise to “Further, four antibiotic efflux resistance mechanisms including major facilitator superfamily (MFS)…”

Response: Thank you. As per the reviewer's suggestions, we have revised the line at line number 291-292.

  1. Line 289 “34,95,232 to 35,09,433 bp” revise to “3,495,232 to 3,509,433 bp”

Response: we have revised the number format as per the suggestions.

  1. Line 322 “spread due their…” revise to “spread due to their…”

Response: Thank you. Now we have revised the line 359.

  1. Line 361 “Several virulence factors have been identified M. morganii genome” revise to “361 “Several virulence factors have been identified in M. morganii genome”

Response: Thank you. Revised accordingly at the line number 408.

Round 2

Reviewer 1 Report

Dear Authors,

Upon reviewing your manuscript, I have a few recommendations to help clarify and enhance your work.

Please revise your usage of the term "orthologous". It appears that in certain instances you've used "orthologous" as a noun, where "orthologs" would be the appropriate term. To clarify, "orthologous" is an adjective typically used in biology to describe genes that have originated from a common ancestor due to speciation, while "orthologs" is the noun form referring to these genes themselves. For instance, you might use "orthologous" to describe a relationship or property (e.g., "orthologous genes"), but "orthologs" would be the correct term when referring to the genes themselves (e.g., "These two genes are orthologs").

Regarding line 282, it seems you intended to refer to the "same strain." In this context, it would be more accurate to use the term "strain" instead of "isolate." This revision should be applied across the entire manuscript, as it is not pre-determined whether two isolates belong to two different strains.

As for Figure 1, I'm curious as to why GCF_018456225 appears more divergent. At times, this divergence can occur due to an incomplete genome sequence. However, given your usage of CHECKM to select high-quality genomes, this doesn't seem to be the case here. To provide further clarity and context, it is common practice to include a supplementary table that presents metadata for each strain. This table would ideally include details such as the name of the isolate, genome length, number of coding sequences (CDS), host, and any other relevant metadata discussed in the paper.

I hope these suggestions are helpful and lead to an even stronger manuscript.

Author Response

Comment: Please revise your usage of the term "orthologous". It appears that in certain instances you've used "orthologous" as a noun, where "orthologs" would be the appropriate term. To clarify, "orthologous" is an adjective typically used in biology to describe genes that have originated from a common ancestor due to speciation, while "orthologs" is the noun form referring to these genes themselves. For instance, you might use "orthologous" to describe a relationship or property (e.g., "orthologous genes"), but "orthologs" would be the correct term when referring to the genes themselves (e.g., "These two genes are orthologs").

Response: We are extremely sorry for these errors are not taken care. This time we have corrected the total manuscript to replace the word "orthologous" to "orthologs".

Comment: Regarding line 282, it seems you intended to refer to the "same strain." In this context, it would be more accurate to use the term "strain" instead of "isolate." This revision should be applied across the entire manuscript, as it is not pre-determined whether two isolates belong to two different strains.

Response: You are very right. We have edited the entire manuscript to correct and replace "isolate" with "strain".

Comment: As for Figure 1, I'm curious as to why GCF_018456225 appears more divergent. At times, this divergence can occur due to an incomplete genome sequence. However, given your usage of CHECKM to select high-quality genomes, this doesn't seem to be the case here. To provide further clarity and context, it is common practice to include a supplementary table that presents metadata for each strain. This table would ideally include details such as the name of the isolate, genome length, number of coding sequences (CDS), host, and any other relevant metadata discussed in the paper.

Response: In the SNP based core tree prepared using all the strains. the GCF_018456225 strain showed nucleotide divergence (Figure 1) which represents the fraction of mismatching nucleotides among the rest of the strains of the core genome alignment (doi.org/10.7554/eLife.65366).

The methodology followed here for the preparation of phylogenetic tree is based on the core tree module of the PGCGAP pipeline, which is subjected to the limitation of providing an automated generated tree and thus, cannot offers the details of SNP analysis of the core genome. The detailed SNP analysis of the core genome, followed by preparation of phylogenetic tree would have provided more precise information about the possible divergence of the strain   GCF_018456225 from the rest, which is beyond the scope of the present study.

In addition, from the ANI analysis, we observed the divergence in ANI values within the phylogenetic tree which indicates the genetic variation, evolutionary divergence, or distinct genomic features between them (supplementary figure S2). Further, the low ANI values of GCF_018456225 genome suggest significant evolutionary divergence due to contamination during genome sequencing.

Reviewer 2 Report

Materials and Methods: Please provide a brief description how SNP calling for the construction of the core SNP tree (Figure 1) was performed. This information should include the parameters used as well as the reference strain used. The authors should consider to include "core SNP calling" under  "Comparative genomics" in Figure S1.

Line 303 (Figure 1) " Tree clustered was shown in different colored" consider revising to "strains associated with the four clusters are delineated by blue, green, light purple, and light brown branches"

Line 308 "7 singlet node" revise to "7 singlet nodes"

Line 308 " can the authors confirm if the 7 singlet nodes comprises 10 phylogenetic branches instead of 11?

General comment

Overall, whilst the content is clear,  proofreading by a native English speaker is required. The grammatical errors distract from the overall quality of the work.

Author Response

Comment: Materials and Methods: Please provide a brief description how SNP calling for the construction of the core SNP tree (Figure 1) was performed. This information should include the parameters used as well as the reference strain used. The authors should consider to include "core SNP calling" under “Comparative genomics" in Figure S1.

Response: We thank the reviewer for pointing this out. As per the suggestion, we have now included a brief description of “core SNP-based phylogenetic tree” in the methodology section.

“The maximum likelihood phylogenetic tree of single-copy core protein was recon-structed using the “CoreTree” module of PGCGAP v1.0.21 and inferences done by plotting the tree using iTol. Briefly, the sequence of single-copy core orthologs were extracted using perl scripts and aligned using MAFFT followed by concatenation of each proteins alignment. Further, the concatenated alignment of each protein was converted into corresponding codon alignment using PAL2NAL v14 followed by calling of core SNPs using SNP-sites. Then, a phylogenetic tree was construed based on best model of evolution using IQ-TREE”

Under “Comparative genomics" in Figure S1 we revised the term “single-copy core proteins tree” to “core SNP-based phylogenetic tree”.

Comment: Line 303 (Figure 1) “Tree clustered was shown in different coloured" consider revising to "strains associated with the four clusters are delineated by blue, green, light purple, and light brown branches".

Response: We agree and have updated the same at line No. 334-335.

Comment: Line 308 "7 singlet node" revise to "7 singlet nodes".

Response: We agree and have updated the same at line No. 340, Page 1.

Comment: Line 308 " can the authors confirm if the 7 singlet nodes comprises 10 phylogenetic branches instead of 11?

Response: From the phylogenetic tree, we found 4 major clusters and 7 singlets which is clearly presented in the figure 1 and in the text also. For the benefit of the readers and not to bring in any confusion, we have rephrased the sentence by eliminating, “which comprises 11 phylogenetic branches” at line number 341.

Comment: Overall, whilst the content is clear, proofreading by a native English speaker is required. The grammatical errors distract from the overall quality of the work.

Response: This is a very good suggestion. We followed the instruction to reform the write up with all possible elimination of grammatical mistakes and syntax errors throughout the manuscript.

Round 3

Reviewer 2 Report

1. Line 204: “were retrieved from the NCBI genome database as on date 30 November 2022” revise to “were retrieved from the NCBI genome database on November 30, 2022”

2. Line 210-211: “To evaluate the genetic relatedness among the genomes, average nucleotide identity (ANI) was calculated using the ‘ANI’ module of PGCGAP v1.0.28 [37] and inferences by plotting the ANI distance matrix” This sentence is incomplete.

3. Line 231-233: “ARGs were identified by BLASTing them against categorized by aligning them with the Comprehensive Antibiotic Resistance Database (CARD) using RGI v5.1.1 [49,50]” This sentence should be rewritten to ensure clarity.

4. Line 239-240: “BLASTn was used to align against the M. morganii serotypes with an identity above 95%” This sentence is incomplete. Do the authors mean to say “BLASTn was used to the align the M. morganii genomes against M. morganii serotypes?”

5. Line 264-266: “This genome has 49.752% genome fraction, 97.01% completeness, and 0.27% contamination.” Consider rewriting to “UM869 had a genome fraction of 49.752% and a genome completeness and contamination level of 97.01% and 0.27% respectively. “

6. Line 270: “virulence factor” revise to “virulence factors”

7. Line 273-274: “The genome also shows 11 resistance genes” revise to “The genome also contains 11 resistance genes”

8. Line 286-287: “The genome contained insertion sequence (IS200G), In36/37 and In6 286 with identity 84%, 99% and 95%, respectively” revise to “The genome also contained the insertion sequences IS200G, In36/37 and In6 with sequence identities of 84%, 99% ….”

9. Line 287-288: “IS200G, the Salmonella-specific insertion sequence contains transposon gene (tnpA) in the genome at position 1945287-1945977 bp 288[55] whereas, In36/37 and In6 were found at 580797-582017 bp and 1049772-1050897 bp respectively (10.1016/0092-8674(83)90552-4)” consider revising to “IS200G is a Salmonella-specific insertion sequence and contains the transposon gene (tnpA) [55]. This gene was located at position 1945287-1945977 bp whereas, In36/37 and In6 were found at position 580797-582017 and 1049772-1050897 bp respectively (10.1016/0092-8674(83)90552-4).”

10. Line 290-292: “These two insertion sequences are of E. coli plasmid origin (AY259086/5 and L06822) bearing the gene HypA (metallo-chaperon), ampR (transcriptional activator) and catA (Chloramphenicol acetyl transferase)” consider revising to “These two insertion sequences are of E. coli plasmid origin (AY259086/5 and L06822) and carry the genes hypA (metallo-chaperon), ampR (transcriptional activator) and catA (Chloramphenicol acetyl transferase).”

11. Line 296:M. morganii starins” revise to “M. morganii strains”

12. Line 297-298:M. morganii genome” revise to “M. morganii genomes”

13. Line 299: “six countries of the world” revise to “six countries”

14. Line 300-301: “The completeness of all the genome was observed in a range of 97.01-100% and contamination was 0-8.66%” revise to “The completeness and contamination levels of all the genomes ranged from 97.01-100% and 0-8.66% respectively.”

15. Line 308: “remaining 79 M. morganii genome” revise to “remaining 79 M. morganii genomes”

16. Line 310-311: “the closest strain based on their genome similarity” revise to “the closest strains based on their genome similarities”

17. Line 319-20: “Because the average ANI percentage was 97.92% among all the genomes, which is greater than cutoff of 95% ANI, all strains belong to the same species” consider revising to “Because the average ANI percentage of all the genomes was 97.92%, which is greater than the ANI cutoff of 95%, all the strains belong to the same species.”

18. Line 320-323: “The ANI tree is divided into three clusters; cluster 1 has 64 strains, clustered 2 has 9 strains and clustered 3 has 6. However, the UM869 strains were (GCA_025398975) closely clustered (99.82% ANI value) with GCF_018474645” consider revising to “The ANI tree (Figure S2) is divided into three clusters, namely 1, 2, and 3 containing 64, 2, and 6 strains respectively. The UM869 strain (GCA_025398975) closely clustered (99.82% ANI value) with GCF_018474645.”

19. ANI tree clusters (line 320-321): Consider labelling the three ANI clusters described in the manuscript. The three clusters cannot be identified by simply looking at the tree unlike Figure 1 where the four clusters described are delineated by color branches.

20. Line 332: “were marked with different colors” revise to “are marked with different colors”.

21. Line 343-344: “We identified a close phylogenetic relationship between UM869 (GCA_025398975), isolated from urine, with GCF_018474645” revise to “We identified a close phylogenetic relationship between UM869 (GCA_025398975), isolated from urine, and GCF_018474645, …….”

22. Line 357: “290050 genes” spell out the number or reword the sentence.

23. Line 374: “2692” spell out the number or reword the sentence.

24. Line 379-380: DNA repair, and recombination proteins revise to “DNA repair and recombination proteins.”

25. Line 386-387: “The highest number (453) of COGs was found in the [S] function unknown category” Rewrite this sentence for clarity.

26. Line 390: “73” revise to “with 73”

27. Line 420: “shown in Table 1” revise to “as shown in Table 1”

28. Line 436-437: “Autotransporters and flagella virulence factor class involved diverse functions such as adhesion, invasion, toxin secretion and host colonization, detected less frequently among the strains” revise to “Autotransporters and flagella virulence factor class involved in diverse functions such as adhesion, invasion, toxin secretion and host colonization, were detected less frequently among the strains”

29. Line 449-450: “The UM869 predicted with a type 8 O-antigen serotype, located in the genome from nucleotide sequences from 3,4,95,232 to 3,5,09,433 bp”. The syntax of this sentence is unclear. It should be rewritten to ensure grammar and readability.

30. Line 461-464: No connection between these sentences and the preceding ones.

31. Line 467: “cassettes is present in 31” revise to “cassettes present in 31”

32. Line 493: “M. morganii “should be italicized

33. Line 497-500: “From Comparative phylogenetic…….” Revise to “Comparative phylogenomic analysis revealed that UM869 genome was closely related to the GCF_018474645 (strain FS112720; isolated from sputum) and GCF_018474565 (strain E89; isolated from secretions) genomes from China.”

34. Line 529: “bacterial adaptation to different environmental conditions by adapting antimicrobial resistance” sentence is unclear; do the authors mean to say “bacterial adaptation to different environmental conditions including antimicrobial resistance?

35. Line 555-556: the first letter for the gene names should be written in lowercase

36. Line 564-565: “The M. morganii genome has identified several virulence factors” revise to “Several virulence factors have been identified in the M. morganii genome including….”

37. Line 588: “strain levels” revise to “strain level”

38. Line 596: “In the study” do the authors mean “In this study”?

39. Line 615: “of its” revise to “of”

40. Line 621: “UM869 strains” revise to “UM869 strain”

Author Response

  1. Line 204: “were retrieved from the NCBI genome database as on date 30 November 2022” revise to “were retrieved from the NCBI genome database on November 30, 2022”

Response: Thank you for the suggestion. Revised accordingly at line no. 204.

  1. Line 210-211:“To evaluate the genetic relatedness among the genomes, average nucleotide identity (ANI) was calculated using the ‘ANI’ module of PGCGAP v1.0.28 [37] and inferences by plotting the ANI distance matrix” This sentence is incomplete.

Response: Thank you for the suggestion. Revised accordingly at line no. 209-211.

  1. Line 231-233:“ARGs were identified by BLASTing them against categorized by aligning them with the Comprehensive Antibiotic Resistance Database (CARD) using RGI v5.1.1 [49,50]” This sentence should be rewritten to ensure clarity.

Response: Thank you for the suggestion. Revised accordingly at line no. 233-235.

  1. Line 239-240:“BLASTn was used to align against the M. morganii serotypes with an identity above 95%” This sentence is incomplete. Do the authors mean to say “BLASTn was used to the align the M. morganii genomes against M. morganii serotypes?”

Response: Yes, now we have corrected in the manuscript at line no. 242-243.

  1. Line 264-266:“This genome has 49.752% genome fraction, 97.01% completeness, and 0.27% contamination.” Consider rewriting to “UM869 had a genome fraction of 49.752% and a genome completeness and contamination level of 97.01% and 0.27% respectively.

Response: Revied accordingly at line no. 267-268.

  1. Line 270:“virulence factor” revise to “virulence factors”

Response:  Revied accordingly at line no. 274.

  1. Line 273-274: “The genome also shows 11 resistance genes” revise to “The genome also contains 11 resistance genes”

Response: Now we have revised the sentence at line no. 278.

  1. Line 286-287:“The genome contained insertion sequence (IS200G), In36/37 and In6 286 with identity 84%, 99% and 95%, respectively” revise to “The genome also contained the insertion sequences IS200G, In36/37 and In6 with sequence identities of 84%, 99% ….”

Response: Now we have revised the sentence at line no. 291-293.

  1. Line 287-288:“IS200G, the Salmonella-specific insertion sequence contains transposon gene (tnpA) in the genome at position 1945287-1945977 bp 288[55] whereas, In36/37 and In6 were found at 580797-582017 bp and 1049772-1050897 bp respectively (10.1016/0092-8674(83)90552-4)” consider revising to “IS200G is a Salmonella-specific insertion sequence and contains the transposon gene (tnpA) [55]. This gene was located at position 1945287-1945977 bp whereas, In36/37 and In6 were found at position 580797-582017 and 1049772-1050897 bp respectively (10.1016/0092-8674(83)90552-4).”

Response: Now we have revised the sentence at line no. 293-299.

  1. Line 290-292:“These two insertion sequences are of E. coli plasmid origin (AY259086/5 and L06822) bearing the gene HypA (metallo-chaperon), ampR (transcriptional activator) and catA (Chloramphenicol acetyl transferase)” consider revising to “These two insertion sequences are of E. coli plasmid origin (AY259086/5 and L06822) and carry the genes hypA (metallo-chaperon), ampR (transcriptional activator) and catA (Chloramphenicol acetyl transferase).”

Response: Now we have revised the sentence at line no. 299-302.

  1. Line 296:M. morganii starins” revise to “M. morganii strains”

Response: Revied accordingly at line no. 307.

  1. Line 297-298:M. morganii genome” revise to “M. morganii genomes”

Response: Revied accordingly at line no. 308-309.

  1. Line 299:“six countries of the world” revise to “six countries”

Response: Revied accordingly at line no. 310.

  1. Line 300-301:“The completeness of all the genome was observed in a range of 97.01-100% and contamination was 0-8.66%” revise to “The completeness and contamination levels of all the genomes ranged from 97.01-100% and 0-8.66% respectively.”

Response: Now we have revised the sentence at line no.  311-313.

  1. Line 308:“remaining 79 M. morganii genome” revise to “remaining 79 M. morganii genomes”

Response: Revied accordingly at line no. 320-321.

  1. Line 310-311:“the closest strain based on their genome similarity” revise to “the closest strains based on their genome similarities”

Response: Revied accordingly at line no. 323-324.

  1. Line 319-20:“Because the average ANI percentage was 97.92% among all the genomes, which is greater than cutoff of 95% ANI, all strains belong to the same species” consider revising to “Because the average ANI percentage of all the genomes was 97.92%, which is greater than the ANI cutoff of 95%, all the strains belong to the same species.”

Response: Now we have revised the sentence at line no.  331-333.

  1. Line 320-323: “The ANI tree is divided into three clusters; cluster 1 has 64 strains, clustered 2 has 9 strains and clustered 3 has 6. However, the UM869 strains were (GCA_025398975) closely clustered (99.82% ANI value) with GCF_018474645” consider revising to “The ANI tree (Figure S2) is divided into three clusters, namely 1, 2, and 3 containing 64, 2, and 6 strains respectively. The UM869 strain (GCA_025398975) closely clustered (99.82% ANI value) with GCF_018474645.”

Response: Now we have revised the sentence at line no.  335-337.

  1. ANI tree clusters (line 320-321):Consider labelling the three ANI clusters described in the manuscript. The three clusters cannot be identified by simply looking at the tree unlike Figure 1 where the four clusters described are delineated by color branches.

Response: Thank you for the suggestion, now we have labelled the cluster numbers in the figure S2.

  1. Line 332:“were marked with different colors” revise to “are marked with different colors”.

Response: Revised at line no. 346.

  1. Line 343-344:“We identified a close phylogenetic relationship between UM869 (GCA_025398975), isolated from urine, with GCF_018474645” revise to “We identified a close phylogenetic relationship between UM869 (GCA_025398975), isolated from urine, and GCF_018474645, …….”

Response: Revised at line no. 357-358.

  1. Line 357:“290050 genes” spell out the number or reword the sentence.

Response: Thank you, we have rewritten the sentence at line no. 372-373.

  1. Line 374:“2692” spell out the number or reword the sentence.

Response: Thank you, we have rewritten the sentence at line no. 383.

  1. Line 379-380:DNA repair, and recombination proteins revise to “DNA repair and recombination proteins.”

Response: Revised at line no. 397.

  1. Line 386-387:“The highest number (453) of COGs was found in the [S] function unknown category” Rewrite this sentence for clarity.

Response: Thank you, we have rewritten the sentence at line no. 406-408.

  1. Line 390:“73” revise to “with 73”

Response: Revised at line no. 411.

  1. Line 420:“shown in Table 1” revise to “as shown in Table 1”

Response: Revised at line no. 449.

  1. Line 436-437:“Autotransporters and flagella virulence factor class involved diverse functions such as adhesion, invasion, toxin secretion and host colonization, detected less frequently among the strains” revise to “Autotransporters and flagella virulence factor class involved in diverse functions such as adhesion, invasion, toxin secretion and host colonization, were detected less frequently among the strains”

Response: Revised at line no. 465-468.

  1. Line 449-450:“The UM869 predicted with a type 8 O-antigen serotype, located in the genome from nucleotide sequences from 3,4,95,232 to 3,5,09,433 bp”. The syntax of this sentence is unclear. It should be rewritten to ensure grammar and readability.

Response: Rewritten at line no. 479-481.

  1. Line 461-464: No connection between these sentences and the preceding ones.

Response: Rewritten the paragraph at line no. 496-500.

  1. Line 467:“cassettes is present in 31” revise to “cassettes present in 31”

Response: Revised at line no. 504.

  1. Line 493:“M. morganii “should be italicized

Response: Revised at line no. 504.

  1. Line 497-500:“From Comparative phylogenetic…….” Revise to “Comparative phylogenomic analysis revealed that UM869 genome was closely related to the GCF_018474645 (strain FS112720; isolated from sputum) and GCF_018474565 (strain E89; isolated from secretions) genomes from China.”

Response: Now we have revised the sentence at line no.  534-537.

  1. Line 529:“bacterial adaptation to different environmental conditions by adapting antimicrobial resistance” sentence is unclear; do the authors mean to say “bacterial adaptation to different environmental conditions including antimicrobial resistance?

Response: Now we have revised the sentence at line no. 565.

  1. Line 555-556:the first letter for the gene names should be written in lowercase

Response: Now we have revised the sentence at line no. 593.

  1. Line 564-565:“The M. morganii genome has identified several virulence factors” revise to “Several virulence factors have been identified in the M. morganii genome including….”

Response: Now we have revised the sentence at line no. 603.

  1. Line 588:“strain levels” revise to “strain level”

Response: Revised at line no. 628.

  1. Line 596:“In the study” do the authors mean “In this study”?

Response: Revised at the line no. 636.

  1. Line 615:“of its” revise to “of”

Response: Revised at the line no. 655.

  1. Line 621:“UM869 strains” revise to “UM869 strain”

Response: Revised at the line no. 661.